# Bushehr Elderly Health (BEH) programme: study protocol and design of musculoskeletal system and cognitive function (stage II)

Gita Shafiee,[1] Afshin Ostovar,[2] Ramin Heshmat,[1] Hossein Darabi,[2] Farshad Sharifi,[3] Alireza Raeisi,[2] Neda Mehrdad,[3] Zhaleh Shadman,[4] Faride Razi,[4] Mohammad Reza Amini,[5] Seyed Masoud Arzaghi,[3] Hamidreza Aghaei Meybodi,[6] Akbar Soltani,[5] Iraj Nabipour,[7] Bagher Larijani[5]

For numbered affiliations see end of article.

**Correspondence to**
Iraj Nabipour;
inabipour@gmail.com and Dr. Bagher Larijani;
emrc@tums.ac.ir

## ABSTRACT

**Introduction** Musculoskeletal disorders and cognitive diseases are prevalent, and they are significant determinants of morbidity and mortality in older adults. The aim of this study is to investigate the prevalence of musculoskeletal and cognitive diseases and their risk factors and also to assess their associations during future follow-ups.

**Methods and analysis** Bushehr Elderly Health (BEH) programme is a population-based prospective cohort study being conducted in Bushehr, a southern province of Iran. A total of 3000 older people aged ≥60 years participated in the first stage from which 2772 were eligible to participate in the second stage, which started after 2.5 years. Data including demographic status, lifestyle factors, general health and medical history, and mental and functional health are collected through a questionnaire. Anthropometric measures, performance tests and muscle strength, blood pressure and and body composition measurements are done. A total 25 cc venous blood is taken, and sera are stored at −80°C for possible future analyses.

**Ethics and dissemination** The study protocol was approved by the ethics committee of Endocrinology and Metabolism Research Institute, affiliated to Tehran University of Medical Science as well as the Research Ethics Committee of Bushehr University of Medical Sciences. A written informed consent was signed by all the participants. The study findings will show the prevalence of musculoskeletal disease, cognitive impairment and their risk factors in an elderly population. The participants will be followed during the study to measure the occurrence outcomes. This study will also have the potential to inform the development of beneficial interventions to improve the management of musculoskeletal and cognitive impairment in Iran and other countries in the Middle East. Our findings will be disseminated via scientific publication as well as presentation to stakeholders, including the patients, clinicians, the public and policymakers, via appropriate avenues.

## Strengths and limitations of this study

► This is the large sample prospective cohort study in Iran focusing on musculoskeletal disorders and cognitive impairment in the elderly as a growing population.
► Future follow-ups in this study will allow for the assessment of many relevant outcomes.
► Collection of the DNA resource for supporting genetic studies in this study will be sought genes associated with musculoskeletal disorders and cognitive impairment.
► The participation rate in this study is high; however, those participants with inadequate physical ability and those elderly living in nursing homes are excluded.

## INTRODUCTION

The world population is ageing at an unprecedented rate.[1] As in most other countries, the proportion of those 60 years and older in Iran was about 8.2% in 2011 and is expected to rise to 10% in 2020.[2] The rise in the number of older people highlights the surging need for understanding the impacts and consequences of this population growth as well as the necessity to conduct research on developing prevention and management strategies for non-communicable diseases (NCDs). Also, to avoid the socioeconomic burden and medical costs, more new effective programmes should be explored. Therefore, health and independence of the elderly have become one of the main priorities of public health systems.[3] In addition, disability is also emerging as an important factor in public health in the world. Many comorbidities occurring with ageing have a great effect on disability. Musculoskeletal disorders are prevalent, and they are significant determinants of morbidity and mortality in older adults. Osteoporosis, characterised by low bone mass and microarchitectural deterioration of bone structure, should be considered as

the most common bone disease. This disease may lead to an increase risk of fragility fractures at the wrist, hip and spine.[4] Considering the increasing life expectancy, the prevalence of osteoporosis is rising dramatically. According to the International Osteoporosis Foundation estimation, currently 200 million women from different parts of the world suffer from the disabling disease.[5] A meta-analysis study have revealed a high prevalence of osteoporosis (12% of males and 19% of females) and osteopenia (33% of males and 40% of females) in Iranian population,[6] adding that some two million of them are at risk of experiencing osteoporosis-related fracture.[7]

Another musculoskeletal disorder is sarcopenia, the age-related decline in muscle mass and function.[8] Decline of muscle mass is started after the age of 50 and an annual loss of muscle mass has been reported to be approximately 1%–2%.[9] Sarcopenia is a major risk factor of falling, functional limitation and disability in the elderly.[10] Functional impairment and physical disability in sarcopenic people are two to three times more likely.[11] Approximately one-third of American women and two-thirds of American men over 60 years suffer from sarcopenia.[12] Disability and loss of independence, falls and fractures are the important complications of osteoporosis and sarcopenia. Therefore, researchers regard such conditions as 'dysmobility syndrome'.[13] In addition, other diseases such as metabolic syndrome, cardiovascular and respiratory diseases are associated with sarcopenia.[14 15] So, the economic burden of sarcopenia is very heavy. Healthcare costs attributable to sarcopenia in the USA in 2000 were estimated at $18.5 billion.[11]

Other important age-related health problems are dementia and cognitive disorders. The incidence of dementia rises with age and can impose a great health and social burden on ageing people. It is estimated that 35.6 million people are living with dementia now and that the number will reach 115.4 million in 2050. In 2010, 57.7% of people with dementia live in developing countries, and this is predicted to increase to 70.5% by 2050.[16] Because dementia is a major cause of disability, it is very important to identify signs and symptoms of dementia at the early stages in order to reduce risk of progressing to an advanced disease and impaired cognitive health. In addition, its significance is likely to increase more than ever keeping in mind that more effective medical treatments are designed and developed to slow the disease process. A subjective memory assessment could potentially be informative of early signs of these diseases, because memory is known to be affected in preclinical dementia and Alzheimer's diseases. Subjective memory impairments have consistently been related to depressive symptoms.[17 18] Nevertheless, studies have indicated that subjective memory impairment is predictive of incident dementia.[19] The risk factors of cognitive impairment and their effect sizes were not recognised yet. However, in the eastern Mediterranean region, there are few prospective studies that have assessed the risk factors of neurocognitive disorders.

Depression is the most frequent mental health problem in the elder population and by the year 2020 has reached the second place in Disability-Adjusted life year(DALY's) ranking.[20] It is a risk factor for other diseases like dementia and hospitalisation.[21] The prevalence of depression in later life is uncertain, which partly is due to the lack of diagnostic criteria for the elderly. WHO estimates that the prevalence of this disorder in the elderly population varies between 10% and 20% depending on cultural situations.[22]

Given the importance of geriatric diseases, several elderly cohort studies are being conducted in developed countries.[23–25] In the Middle East, there are a few large longitudinal studies with elderly populations.[26] Bushehr Elderly Health (BEH) programme is a population-based prospective cohort study is being conducted in Bushehr, a southern province of Iran. The main aim of its first stage was to investigate the prevalence of cardiovascular risk factors and their association to major adverse cardiovascular outcomes.[27] In this stage, the study has been designed and implemented to determine the prevalence of musculoskeletal disorders, cognitive impairment and their risk factors, as well as investigation of falling, fractures, dementia, poor mobility and functional dependence in each follow-up assessment. Also, we will investigate a number of possible molecular mechanisms linking to geriatric diseases.

## BEH PROGRAMME
### Study design
The methodology of the BEH programme is previously described elsewhere,[27] and here, its summary and the methods related to the current phase are explained. The BEH programme is a prospective cohort study aimed at investigating the prevalence of NCDs and its associated risk factors. Baseline measurements of the first stage were implemented from March 2013 to October 2014. The second stage has begun from October 2015, and data recollection is designed to be in 2.5-year intervals.

Four objectives relating to musculoskeletal disorders and cognitive impairment are identified as follows:
► to determine the prevalence dementia and cognitive impairment and their risk factors among elderly Iranians and assessment of associations with their outcomes during future follow-ups
► to estimate the prevalence musculoskeletal disorders (osteoporosis and sarcopenia) and the risk factors among elderly people and assessment of associations with their outcomes during future follow-ups
► to establish a uniform database for performing subsequent studies of the natural history of musculoskeletal disorders and cognitive impairment for planning and evaluating interventions
► to collect the DNA resource for supporting genetic studies that have sought genes associated with musculoskeletal disorders and cognitive impairment.

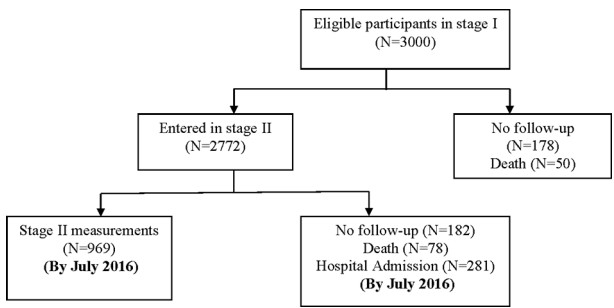

**Figure 1** Flow chart of enrolments in stage II.

The study also aims to act as a core resource and provides a framework to support specific substudies about healthy ageing at the national and international levels.

## Study population

The target population of this cohort study was all participants aged 60 years residing in the city of Bushehr, the centre of a southern province of Iran. Based on the information available from the Bushehr District Health Centre, this population was estimated to be around 10 000. A total of 3000 individuals were selected using a multistage cluster random sampling method in the first stage of sampling 2.5 years prior to the beginning of the second stage.

This sample size is enough for estimation of dementia prevalence with a rate of 8% ((6.5%–9%)%) and with a calculated power of 83% STATA (Release V.12).[28]

The participants had been selected from 75 neighbourhoods defined by the municipality. The number of participants recruited from each neighbourhood was proportional to the number of households reported from the last census (about 2 years prior to the stage I). The participation rate for the first stage was about 91%. The participants and an accompanied person (usually a relative) signed the informed consent. None of those who came to the recruiting centre declined to sign the form.

Older people who were in care homes were excluded from the study. Details of inclusion and exclusion criteria have been described previously,[27] and figure 1 shows the flow chart of enrolments in stage II.

## Outcome measurements

Every participant will be followed up for any medical event (fracture, falling and death) annually by telephone call. Each participant has been asked for any medical conditions by a trained nurse and, if necessary, a trained physician will investigate their hospital records. In the case of mortality, data are collected from hospital or death certificate by local physicians and are verified by death registry system in the district health centre. The collected data are then evaluated by an outcome committee consisted of principal investigator, internist, endocrinologist, epidemiologist and physician who collected outcome data.

**Table 1** Section and topics of the questionnaire in the Bushehr Elderly Health programme study (stage II)

| Type of data | Parts |
| --- | --- |
| Demographic data | Personal information |
| | National identification number |
| | Age/sex |
| | Marital status |
| | Contact information |
| Lifestyle factors | Smoking (current and history) |
| | Alcohol consumption |
| | Drug abuse |
| | Physical activity |
| | Nutritional assessment |
| General health and medical history | Quality of life |
| | Medical history |
| | Musculoskeletal status (current and history) |
| | Women's health |
| Mental and functional Health | Cognitive function |
| | Depression |
| | Functional assessment |
| | Activities of daily living |
| Medication use | Administrated by physician |
| | Over-the-counter drugs |
| | Supplements |

## Data collection

In order to collect the necessary data, a questionnaire was developed by expert groups and was tested in a pilot study to identify problems with the parts and to estimate their time of administration. The majority of questions were taken from previous validated questionnaires.[29–32]

All participants invited to the BEH programme unit were referred to trained nurses, after signing informed written consent. The participants are interviewed by trained nurses to complete a 162-item questionnaire regarding sociodemographic data, lifestyle factors, general health and medical history, mental and functional health, and medication use. Table 1 presents the parts of the core questionnaire.

Physical activity level in 24 hours of sports, work and leisure time on an average weekday is assessed using a valid self-report questionnaire,[33] and the intensity of every specific activity is expressed in metabolic equivalents. This instrument has been validated and used for elderlies[34–36] and also, it has been validated among Iranian adolescents for Farsi language.[32]

Dietary assessment is performed by a 24-hour dietary recall to obtain detailed information about all foods and beverages consumed in the past 24 hours. The total amount of each specific food and beverage consumed is captured by expert nutritionist. Standard reference tables

are used to convert household portions to grams. Data are entered into the nutritionist IV package, modified for Iranian foods to obtain daily energy, nutrient intakes and servings of foods consumed. For mixed dishes, food groups and nutrients are calculated according to their ingredients.

Also, for evaluating the risk of malnutrition in elderly people, we use the Mini Nutritional Assessment. This practical assessment tool contains: anthropometric measurements, global assessment, dietary questionnaire and a subjective assessment that enables a subject to be categorised as normal (adequate nutrition), borderline (at risk of malnutrition) or undernutrition.[37]

Mental and functional health assessments are performed by using the MMSE, Mini-Cog, functional assessment staging test (FAST), the 12-item Short Form Health Survey (SF-12), the Patient Health Questionnaire-9 (PHQ-9), activities of daily living (ADL) and instrumental activities of daily living (IADL) for all participants. They have already been validated and translated for use in primary care in Iran[29 30 38 39]:

► MMSE: a brief quantitative tool that can be used to assess mental status in adults. It is sensitive to screen for cognitive impairment, to estimate the severity of disease, to identify early changes of cognitive status and to evaluate response to treatment. This tool is an 11-question measure that tests five areas of cognitive function, including orientation, registration, attention and calculation, recall and language.[40]

► Mini-Cog test: another brief screen test for cognitive impairment is the Mini-Cog. It takes approximately 3 min to administer. Its components, including three-item recall and clock drawing are combined to discriminate dementia. It has minimal language content, which reduces cultural and educational bias.[41]

► FAST: the evaluation of changes in functional performance and ADLs skills in elderly individuals, especially patient with dementia disease, is performed by FAST. The FAST scale is composed of seven major levels of functioning (from normal adult to severe cognitive decline); levels 6 and 7 are additionally divided into substages (11 total).[42]

► PHQ: it is a self-report version of the Primary Care Evaluation of Mental Disorders diagnostic instrument for common mental disorders.[43] The PHQ-9 has the potential of being a dual-purpose instrument that, with the same nine items, can establish depressive disorder diagnoses as well as grade depressive symptom severity.[44]

► SF-12: Thehealth-related quality of life (QoL) that describes the degree of general physical health status and mental health distress, is performed by the SF-12. SF-12 measures eight health domains, including physical functioning, role physical, bodily pain, general health, vitality, social functioning, role emotional and mental health. It provides glimpses into mental and physical functioning and overall QoL.[45]

---

**Box 1  Physical examinations in Bushehr Elderly Health programme study (stage II)**

**Anthropometric measures**
► Height and weight
► Waist circumference
► Hip circumference
► Neck circumference
► Upper arm circumference
► Forearm circumference
► Mid-thigh circumference
► Calf circumference

**Performance-based tests**
► Short physical performance battery (SPPB)
  ► Time to perform five chair stands
  ► Walking 2.44 m at usual gait speed
  ► Balance test
    ► Side-by-side standing position
    ► Full tandem position
    ► Semitandem position
► Walking 4.57 m at usual gait speed

**Muscle strength**
► Grip strength

**Blood pressure measurements**

**Body composition measurements**
► Fat mass
  ► Muscle mass
  ► Bone mineral density

---

► ADLs and IADL: they are as part of an older person's functional assessment. ADLs are routine tasks to refer to people's daily self-care activities. ADLs include: eating, bathing, dressing, toileting, transferring (walking) and continence. An individual's ability to perform ADLs is important for determining what type of long-term care (eg, nursing home care or home care) and coverage the individual needs.

IADLs are the complex skills needed to successfully live independently. These skills are cooking, driving, using the telephone or computer, shopping, keeping track of finances and managing medication. Together, ADLs and IADLs represent the skills that usually make people able to manage their own life as independent adults.[46]

### Clinical examination
#### Anthropometric measurements

Anthropometric measurements (Box 1) are taken with shoes removed and the participants wearing light clothing. Height and weight are measured with a fixed stadiometer and a digital scale according to the standard protocol. A flexible, circumference measuring tape is used to measure waist (WC) and hip circumferences. WC should be measured at a point midway between the iliac crest and the lowest rib in standing position, and hip circumference should be measured as the maximal circumference over the buttocks.

Neck circumference is measured in the midway of the neck, between midcervical spine and mid anterior neck, just below, with the subjects standing upright. Upper arm circumference is measured at the midpoint between the

olecranon process and the acromion of right arm; also, forearm circumference measurement is done from the widest level while the arm is hanging freely at the side.

Midthigh circumference is measured at a midpoint between trochanterion (top of the thigh bone, femur) and tibiale laterale (top of the tibia bone) of right thigh. Calf circumference measurement is measured at the widest level while the participant is standing upright. All measurements are read to the nearest 0.1 cm.

### Performance-based tests and muscle strength

To test muscle function, hand grip strength (muscle strength), 4.57-m usual gait speed (physical performance) and the SPPB are measured.[47] Handgrip strength is measured in both hands using a digital dynamometer, and each hand is measured three times. The main hand is first checked, after which participants are allowed to rest for 15 s; then, the other hand is checked. A 4.57 m walk at the subject's normal pace is timed twice, and the time of the faster of the two walks is used for usual gait speed.

SPPB is a group of measures that combines the results of the gait speed, chair stand and balance tests to measure physical performance in older adults. To test 'the chair stand', participants are asked to keep their arms folded across their chest one time. If successful, participants were asked to stand up and sit down five times and are timed from the initial sitting to final standing position. 'Balance tests' include full tandem, semitandem and side-by-side stands. For each stand, the interviewer supports one arm while participants position their feet, asks if they are ready, then releases the support and begins timing. When 10 s have elapsed or participants move their feet, the timer is stopped. To test 'gait speed', at 2.44-metre usual gait speed is measured.

### Blood pressure measurements

Two measurements of systolic blood pressure (SBP) and diastolic blood pressure (DBP) are performed using a standardised mercury sphygmomanometer on the right arm after a 15 min rest in a sitting position; the first and fifth Korotkoff sounds are taken as SBP and DBP, respectively. The average of the two measurements is considered as the participant's blood pressure.

### Body composition measurements

Body composition for each participant is measured using dual x-ray absorptiometry (DXA, Discovery WI, Hologic, Bedford, Virginia, USA). This device can measure fat mass, muscle mass, head, trunk and extremities separately with minimal radiation exposure. According to DXA results, we will calculate the appendicular skeletal muscle mass for each participant as the sum of upper and lower limb muscle mass. Also, the bone mineral density of the lumbar spine (L1-L4) and total hip are measured in a correct position. All scans are performed by a trained operator on the same day. The time, which is required for the evaluation of each person, will be 20 min.

**Table 2** Laboratory tests of Bushehr Elderly Health programme study (phase II)

| Laboratory tests | Method of measurement |
|---|---|
| CBC | Automated haematology analyser, Medonic CA620 (Menarini Diagnostics Srl, Florence, Italy) |
| Fasting blood sugar | Enzymes (glucose oxidase) colorimetric method using a commercial kit (Pars Azmun, Karaj, Iran) |
| Lipid profile Total cholesterol | Enzymatic (CHOD-PAP) colorimetric method using a commercial kit (Pars Azmun) |
| LDL-C | Enzymatic (CHE& CHO) colorimetric method using a commercial kit (Pars Azmun) |
| HDL-C | Enzymatic (CHE& CHO) colorimetric method using a commercial kit (Pars Azmun) |
| Triglycerides | Enzymatic (GPO-PAP) colorimetric method using a commercial kit (Pars Azmun) |
| Calcium | Enzymatic colorimetric estimation with o-cresolphthalein complex method using a commercial kit (Pars Azmun) |
| Phosphorus | Enzymatic UV method with phosphomolybdate using a commercial kit (Pars Azmun) |
| ALK-p | Photometric method using a commercial kit (Pars Azmun) |
| Urea | Enzymatic (urease) UV method using a commercial kit (Pars Azmun) |
| Creatinin | Jaffe's method, using a Commercial Kit (Pars Azmun) |
| Uric acid | Colorimetry and uricase method using uric acid TOOS kit (Pars Azmun) |
| HbA1C | Boranate affinity method using a CERA-STAT system (*CERAGEM MEDISYS*,chungcheongnam-do, Korea) |

ALK-p, alkaline phosphatase; CBC, complete blood count; CHE, cholesterol esterase; CHO, cholesterol oxidase; CHOD-PAP, cholesterol oxidase phenol aminoantipyrine; GPO-PAP, glycerol-3-phosphate oxidase phenol aminoantipyrine; HbA1C, haemoglobin A1C; HDL, high-density lipoprotein; LDL, low-density lipoprotein; UV, ultraviolet.

### Biochemical measurements

An overnight fasting venous blood sample is obtained for every participant for biochemical measurements. A total of 25 cc of whole blood is collected by a trained nurse. Table 2 presents laboratory tests performed at baseline and their methods of measurement. Sera are also separated and stored at –80°C for possible future analyses.

DNA will also be extracted from whole peripheral blood stored in EDTA tubes using standard methods for future genome-wide association studies and epigenome-wide association studies.

## STATISTICAL ANALYSES

All data will be analysed using SPSS (V.21) and STATA (Release V.12). Normality of variables would be assessed using visual inspection of histograms and the measures of skewness and kurtosis as well as Kolmogorov-Smirnov and Shapiro-Wilks tests. Data will be presented as means±SD, medians and IQR for continuously distributed variables and frequencies and percentages of categorical variables. For comparing descriptive results of the population, prevalence or incidence rates will be standardised based on country population at the same age–sex groups.

Appropriate parametric and non-parametric statistical methods will be performed for data analyses. Student's t-test or analysis of variance and their corresponding non-parametric methods (Mann-Whitney and Kruskal-Wallis, respectively) will be used to compare continuous variables between the different groups. Pearson and Spearman correlation coefficients will be performed to assess the relationship between continuous variables. Pearson $\chi^2$ will be used to assess the association of categorical variables. Risk ratio, rate ratio, OR and their 95% CIs will be used to show the association of binary outcomes and corresponding risk factors. Survival analysis will be used for time to event data. The survival rate and HR will be used to compare between groups. Appropriate multivariable analyses (linear regression, logistic regression, Poisson regression, analysis of covariance methods, Cox proportional hazard models and so on) will be used for the purpose of investigating associations after controlling for potential confounders.

Missing data will be treated by performing appropriate methods, including replacing missing values with the mean of the observed values, the last measured values, sensitivity analysis of the best and worst scenarios and multiple imputation analyses.[48]

The level of p value ≤0.05 will be considered to indicate statistical significance.

## ETHICS

The study protocol was approved by the ethics committee of Endocrinology and Metabolism Research Institute, affiliated to Tehran University of Medical Science as well as the Research Ethics Committee of Bushehr University of Medical Sciences. A written in forming consent was signed by all the participants, including the permission to use secondary data such as registration data for health service purposes, before enrolment in the study

## STRENGTHS AND LIMITATIONS

There are several strengths of this study. First, this is the cohort study of older people for a longer follow-up time. Therefore, this study may assist in determining causal associations between predicted determinants of health and outcomes and could be used to obtain a multifaceted understanding of geriatric health over time. Also, this study enjoys a large sample size of elderly participants to assess musculoskeletal and cognitive diseases. The face-to-face interview is useful in getting exact information, especially among the older people who may not immediately understand or respond to the questionnaire. Lastly, our multidisciplinary investigative team consists of experts in geriatrics, psychology, endocrinology, epidemiology and genetics that may be helpful in developing a unique research resource for the understanding of geriatric health.

Attrition bias caused by selection bias as well as recall problems of the elderly people from the very first stage may be considered as the limitations of the current study. Long follow-up and case ascertainment problems should also be mentioned as other limitations. Another limitation of this study is to enter only those participants with adequate physical ability and the exclusion of those elderly living in nursing homes, which has led to selection bias. The self-reported physical activity scale tool has been validated in Farsi language only among Iranian adolescents and not especially in elderlies. However, it has been validated for elderlies in other studies.

## CONCLUSION

This study protocol describes the design and methods used in the stage II of BEH programme. In this study, we focused on the epidemiological characteristics of geriatric disorders, the relationship between socioeconomic factors and health and QoL in addition to elucidating biological risk factors of musculoskeletal and cognitive diseases, and assessment of associations during future follow ups in elderly persons in Bushehr, Iran. The follow-up assessments for each stage will be carried out every 5 years for three consecutive periods (a total of 15 years of follow-up), in which all assessments will be repeated with comparable methods and modalities.

The study findings will improve our understanding of disease prevalence in the elderly of our country and will have the potential to inform the development of beneficial interventions to improve the management of musculoskeletal diseases and cognitive impairment in Iran and other countries in the Middle East and North Africa region.

**Author affiliations**
[1]Chronic Diseases Research Center, Endocrinology and Metabolism Population Sciences Institute, Tehran University of Medical Sciences, Tehran, Iran
[2]The Persian Gulf Tropical Medicine Research Center, Bushehr University of Medical Sciences, Bushehr, Iran
[3]Elderly Health Research Center, Endocrinology and Metabolism Population Sciences Institute, Tehran University of Medical Sciences, Tehran, Iran
[4]Diabetes Research Center, Endocrinology and Metabolism Clinical Sciences Institute, Tehran University of Medical Sciences, Tehran, Iran
[5]Endocrinology and Metabolism Research Center, Endocrinology and Metabolism Clinical Sciences Institute, Tehran University of Medical Sciences, Tehran, Iran
[6]Osteoporosis Research Center, Endocrinology and Metabolism Clinical Sciences Institute, Tehran University of Medical Sciences, Tehran, Iran
[7]The Persian Gulf Marine Biotechnology Research Center, The Persian Gulf Biomedical Sciences Research Institute, Bushehr University of Medical Sciences, Bushehr, Iran

**Acknowledgements** The authors are grateful to the staff of both research centres at Bushehr University of Medical Sciences (BPUMS) and Tehran University

of Medical Sciences (TUMS) for their commitment to the study's protocol and objectives. They are also indebted to all participants who accepted the invitation and patiently underwent exhausting measurements and examinations.

**Contributors** GS drafted the manuscript, participated in questionnaire development study design and staff training. AO and RH participated in the study design, questionnaire development and conduction and reviewed the manuscript. HD, FS, AR, FR, S.M.A and ZS participated in questionnaire development, study design and staff training and reviewed the manuscript. NM, MRA, HAM and AS participated in questionnaire development. IN and BL conceived the study, helped draft the manuscript and participated in the study design and conduction.

**Competing interests** None declared.

**Patient consent** Obtained.

**Provenance and peer review** Not commissioned; externally peer reviewed.

**Data sharing statement** Data will be available for interested researchers upon their request via our website or through direct email to the executive manager AO ( a.ostovar@bpums.ac.ir; afshinostovar@gmail.com).

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
