## [Reviewer comments · BMJ Open]

ARTICLE DETAILS

TITLE (PROVISIONAL)	Bushehr Elderly Health (BEH) program: Study protocol and design of Musculoskeletal System and Cognitive Function (Stage II)
AUTHORS	Larijani, Bagher; Shafiee, Gita; Ostovar, Afshin; Heshmat, Ramin; Darabi, Hossein; Sharifi, Farshad; Raeisi, Alireza; Mehrdad, Neda; Shadman, Zhaleh; Razi, Faride; Amini, Mohammad Reza; Arzaghi, Seyed Masoud; Aghaei Meybodi, Hamidreza; Soltani, Akbar; Nabipour, Iraj

VERSION 1 - REVIEW

REVIEWER	John Gladman University of Nottingham, UK
REVIEW RETURNED	12-Aug-2016

GENERAL COMMENTS	Thank you for asking me to review this protocol for this study that is in progress. Longitudinal cohort studies such as this are of enormous value and much needed to explore the emerging epidemiology of ageing. Publication of the methods will be useful for those doing similar work and to anticipate the findings. My comments are as follows. The written English needs improvement, with respect to spelling, use of capitals (e.g. sarcopenia), punctuation and spacing, some abbreviations (NCD for example) and so forth. The errors are numerous and I have not proofed the text for these. There are also a few bits that don't quite make sense such as on page 6 lines 17-18 it is stated that "Because dementia ... major cause of disability ... early diagnosis is important" . This not a logical statement, even though the first and second clauses are true. With regards to the actual content: In the Abstract, I would like to see the numbers of people recruited in phase 1 and eligible for phase 2 mentioned. In the Methods, I would like to see some mention of how missing data will be handled. Also, some estimation of the power the study will have to detect incident dementia and disability. Also a bit more on the analysis plan: one of the potential problems of these studies is the risk of data dredging rather than pre-planned hypothesis testing. Overall, I would like some clarification: • when were the potential participants eligible (i.e. when did they have to reach ≥ 60)?• can a brief sentence explain how people were recruited – could the methods under represent people with disability or cognitive impairment (the need for signed informed consent could lead to
--

	under representation of people with cognitive impairment)? I note at the end of the paper it is stated that people in care homes were excluded, which is likely to be important to state in the Method and not only in the limitation section.  • Page 13 line 18 says follow up is every 5 years: elsewhere 2.5 years is mentioned • Ascertainment of falls by an annual question is not very accurate, and deserves a specific mention in the limitations
--	--

REVIEWER	Somnath Chatterji World Health Organization Switzerland
REVIEW RETURNED	21-Oct-2016

GENERAL COMMENTS	This paper describes the second phase that is ongoing in the Bushehr cohort in Iran - a cohort that has been under study since 2013 but with added objectives in the current phase. While the authors reference the original publication, it would help readers to get a sense of how representative of the older adult population in Iran is the cohort. The authors could provide a little more detail on the age-sex structure of the cohort and how this compares to Iran as a whole. This would help understand how generalizable the results are likely to be. It is unclear if the measure of physical activity chosen (the reference 31 is to a study among adolescents) has been tested in older adults and if not why this measure has been chosen over other internationally used measures such as the IPAQ or GPAQ. I tried finding the full text of the cited reference but couldn't find it on the journal's website. The authors mention the objective of collecting DNA but this is not mentioned later on in the details of the methods. Is the DNA going to be isolated from the venous blood samples? When will this be done? How much DNA will be stored? Is there a plan for what genomic analyses are envisaged at the present moment? Finally, there is no mention of whether the anonymized data will be made available to other researchers from a data archive or upon request to the lead investigators (upon what conditions?) or not at all. It would help readers to know this.
--

VERSION 1 – AUTHOR RESPONSE

Reviewer 1

1. Comment:

The written English needs improvement, with respect to spelling, use of capitals (e.g. sarcopenia), punctuation and spacing, some abbreviations (NCD for example) and so forth. The errors are numerous and I have not proofed the text for these.

- Author response:

Thanks for your comment. We corrected the manuscript.

2. Comment:

There are also a few bits that don't quite make sense such as on page 6 lines 17-18 it is stated that

“Because dementia ... major cause of disability ... early diagnosis is important”. This not a logical statement, even though the first and second clauses are true.

- Author response:

We corrected the sentence.

3. Comment:

In the Abstract, I would like to see the numbers of people recruited in phase 1 and eligible for phase 2 mentioned

- Author response:

We added the number of people recruited in phase 1 and eligible for phase 2 in “Abstract” part.

4. Comment:

In the Methods, I would like to see some mention of how missing data will be handled.

- Author response:

A paragraph was added to the end of statistical analysis section.

5. Comment:

Also, some estimation of the power of the study will have to detect incident dementia and disability.

Also a bit more on the analysis plan: one of the potential problems of these studies is the risk of data dredging rather than pre-planned hypothesis testing

- Author response:

A sentence was added to the “study population” section.

Based on pre-defined objectives in the protocol, prevalence of risk factors and health-related outcomes will be assessed and the hypothesis of study will be analyzed.

6. Comment:

Overall, I would like some clarification:

- when were the potential participants eligible (i.e. when did they have to reach ≥ 60)?

- Author response:

We have made it clear that all the participants aged 60 years and older in stage I re-invited for second stage. List of other inclusion/exclusion criteria is available in cohort profile paper cited in this manuscript.

7. Comment:

Can a brief sentence explain how people were recruited – could the methods under represent people with disability or cognitive impairment (the need for signed informed consent could lead to under representation of people with cognitive impairment)?

- Author response:

A paragraph was added to the “study population” section. However, there is still risk of underestimation of the rate as we had excluded those with severe mental problems.

8. Comment:

I note at the end of the paper it is stated that people in care homes were excluded, which is likely to be important to state in the Method and not only in the limitation section.

- Author response:

A sentence was added to the study population section.

9. Comment:

Page 13 line 18 says follow up is every 5 years: elsewhere 2.5 years is mentioned.

- Author response:

The follow-up assessments for each stage will be carried out every 5 years for three consecutive periods (a total of 15 years of follow-up). But, people aged ≥ 60 were participated in the first stage, were eligible to participate in the second stage started after 2.5 years.

10. Comment:

Ascertainment of falls by an annual question is not very accurate, and deserves a specific mention in the limitations

- Author response:

We have made it clear in outcome measurement section that we will ask if the participant have experienced fall or fracture during the year before. Then, further information will be gathered by a general physician from medical records.

Reviewer2

1. Comment:

While the authors reference the original publication, it would help readers to get a sense of how representative of the older adult population in Iran is the cohort. The authors could provide a little more detail on the age-sex structure of the cohort and how this compares to Iran as a whole. This would help understand how generalizable the results are likely to be.

- Author response:

A sentence was added to the Statistical analyses section.

2. Comment:

It is unclear if the measure of physical activity chosen (the reference 31 is to a study among adolescents) has been tested in older adults and if not why this measure has been chosen over other internationally used measures such as the IPAQ or GPAQ. I tried finding the full text of the cited reference but couldn't find it on the journal's website.

- Author response:

We added a reference among adults (the reference 33).

Physical activity questionnaires could fail in estimating particularly non-vigorous physical activity. They have a focus on volitional type exercise (i.e. biking, jogging, and walking), while not capturing the activities of daily living and low to moderate intensity movements. Energy expenditure estimates from these data are not recommended. On the other hand, despite objective tools should be the measurement of choice to assess PA level, self-reported questionnaires remain valid, and have many advantages. i.e. low costs. These kinds of recall are designed and validated for different age groups and provide value and important information, mainly about physical activity pattern.

3. Comment:

The authors mention the objective of collecting DNA but this is not mentioned later on in the details of the methods. Is the DNA going to be isolated from the venous blood samples? When will this be done? How much DNA will be stored? Is there a plan for what genomic analyses are envisaged at the present moment?

- Author response:

A paragraph was added to the Biochemical measurement section.

For your kind information we have recently joined an international consortium (GSA Consortium) for genotyping using new Illumina array.

4. Comment:

Finally, there is no mention of whether the anonymized data will be made available to other researchers from a data archive or upon request to the lead investigators (upon what conditions?) or not at all. It would help readers to know this.

- Author response:

Data sharing statement section was added to the end of the paper.

VERSION 2 – REVIEW

REVIEWER	Somnath Chatterji World Health Organization
-----------------	--

	Switzerland
REVIEW RETURNED	03-Feb-2017

GENERAL COMMENTS	While the authors respond to some of the comments, some are inadequately responded to: 1. With regard to how the cohort population compares to the rest of Iran there is no statement though the authors state they have added a statement in the statistical analysis. This has implications for understanding how generalizable these results are. 2. With regard to physical activity, in their response they state that the instrument they are using should not be used to estimate energy expenditure and yet in the paper they still state " The intensity of every specific activity is expressed in metabolic equivalents (METs), which allows for the estimation of energy expenditure of specific physical activities". Additionally, reference 33 does not mention the specific instrument they are going to use in this cohort. My earlier comment the first time around was that the authors had mentioned they were going to use an instrument that has been used with adolescents and it was unclear how valid this instrument was for older adults. The authors need to specify which instrument they are going to use and justify its use or at least acknowledge the limitations of their choice.
---

VERSION 2 – AUTHOR RESPONSE

Reviewer: 2

1. Comment:

- Please state any competing interests or state 'None declared': None declared.

- Author response:

We added state 'None declared'

2. Comment:

With regard to how the cohort population compares to the rest of Iran there is no statement though the authors state they have added a statement in the statistical analysis. This has implications for understanding how generalizable these results are.

- Author response:

- We added a statement in statistical analysis: For comparing descriptive results of the population, prevalence or incidence rates will be standardized based on country population at the same age- sex groups.

3. Comment:

- With regard to physical activity, in their response they state that the instrument they are using should not be used to estimate energy expenditure and yet in the paper they still state " The intensity of every specific activity is expressed in metabolic equivalents (METs), which allows for the estimation of energy expenditure of specific physical activities". Additionally, reference 33 does not mention the specific instrument they are going to use in this cohort. My earlier comment the first time around was

that the authors had mentioned they were going to use an instrument that has been used with adolescents and it was unclear how valid this instrument was for older adults. The authors need to specify which instrument they are going to use and justify its use or at least acknowledge the limitations of their choice.

- Author response:

We corrected the manuscript and added a statement about this issue in the Methods and limitation:
Methods:

This instrument has been validated and used for elderlies (34-36) and also, it has been validated among Iranian adolescents for Farsi language (32).

Limitations:

The self-reported physical activity scale tool has been validated in Farsi language only among Iranian adolescents and not especially in elderlies. However, it has been validated for elderlies in other studies.